# Histopathological Analysis for Detecting Lung and Colon Cancer Malignancies Using Hybrid Systems with Fused Features

**DOI:** 10.3390/bioengineering10030383

**Published:** 2023-03-21

**Authors:** Mohammed Al-Jabbar, Mohammed Alshahrani, Ebrahim Mohammed Senan, Ibrahim Abdulrab Ahmed

**Affiliations:** 1Computer Department, Applied College, Najran University, Najran 66462, Saudi Arabia; 2Department of Artificial Intelligence, Faculty of Computer Science and Information Technology, Alrazi University, Sana’a, Yemen

**Keywords:** VGG-19, GoogLeNet, lung and colon cancer, FCH, ANN, GLCM, PCA, LBP, DWT

## Abstract

Lung and colon cancer are among humanity’s most common and deadly cancers. In 2020, there were 4.19 million people diagnosed with lung and colon cancer, and more than 2.7 million died worldwide. Some people develop lung and colon cancer simultaneously due to smoking which causes lung cancer, leading to an abnormal diet, which also causes colon cancer. There are many techniques for diagnosing lung and colon cancer, most notably the biopsy technique and its analysis in laboratories. Due to the scarcity of health centers and medical staff, especially in developing countries. Moreover, manual diagnosis takes a long time and is subject to differing opinions of doctors. Thus, artificial intelligence techniques solve these challenges. In this study, three strategies were developed, each with two systems for early diagnosis of histological images of the LC25000 dataset. Histological images have been improved, and the contrast of affected areas has been increased. The GoogLeNet and VGG-19 models of all systems produced high dimensional features, so redundant and unnecessary features were removed to reduce high dimensionality and retain essential features by the PCA method. The first strategy for diagnosing the histological images of the LC25000 dataset by ANN uses crucial features of GoogLeNet and VGG-19 models separately. The second strategy uses ANN with the combined features of GoogLeNet and VGG-19. One system reduced dimensions and combined, while the other combined high features and then reduced high dimensions. The third strategy uses ANN with fusion features of CNN models (GoogLeNet and VGG-19) and handcrafted features. With the fusion features of VGG-19 and handcrafted features, the ANN reached a sensitivity of 99.85%, a precision of 100%, an accuracy of 99.64%, a specificity of 100%, and an AUC of 99.86%.

## 1. Introduction

Cancer is the second leading cause of death. In 2020, more than 19 million people diagnosed with cancer were reported, and about 10 million people died worldwide [1]. The human body contains trillions of cells that grow and multiply through division. Some cells are damaged or reach a certain age, die, and are replaced by normal cells. If not replaced by normal cells, the damaged cells grow, multiply, and form benign or malignant tumors [2]. Cells growing slowly and not damaging the surrounding tissues are benign tumors. In contrast, the cells that grow abnormally and quickly and damage the surrounding tissues are malignant tumors. Cancer cells infect many parts of the human body. Lung and colon cancers are among the most common cancers affecting males and females equally. In 2020, the number of people diagnosed with lung and colon cancer reached 4.19 million, and more than 2.7 million died worldwide. There are many behavioral causes of cancer, such as cigarette use, increased body mass, and alcohol abuse, along with exposure to ultraviolet rays, radiation, and some biological and other factors that cause cancer [3]. There are cases of simultaneous occurrence of the two types of lung and colon cancer by 17%. According to the study, smoking is a cause of breast cancer, which causes an abnormal diet, which also causes colon cancer [4]. The patient does not show any signs of the disease in the early stages or shows few signs, but when symptoms appear, the tumor is in the advanced stages, and it is too late. It is difficult to detect lung and colon cancer in its early stages without comprehensive examinations such as CT scans, MRI, ultrasound and tissue examination [5,6]. Therefore, smokers, those who are overweight, and those with a genetic history should have regular check-ups. Screening techniques are expensive, and many people on low incomes cannot afford the costs. According to the World Health Organization, 70% of cancer deaths are in low- and middle-income developing countries. Therefore, these countries must be supported to build hospitals with fully-equipped and free-of-charge diagnostic laboratories and train medically qualified workers to carry out appropriate diagnostic procedures. Moreover, diagnosing cancer cases takes a long time and is subject to differing opinions of doctors, especially in the early stages. A different field of health care can overcome these challenges. Artificial intelligence techniques are used in the healthcare field, such as early diagnosis of biomedical images, predictions of diseases, health disasters, and others [7]. Deep learning techniques have the ability to analyze data from high-dimensional images, videos [8], and anatomical representations [9]. Moreover, deep learning techniques extract hidden features and characteristics from medical images that cannot be seen with the naked eye for the early detection of cancers and discrimination between their stages. Because of the similar characteristics of the abnormal cells in the early stages, in this study, several hybrid systems were developed with features extracted by mixed methods. The study aims to extract the features of more than one algorithm and combine them with deep learning features. Thus, each type of cancer is represented by strong features that distinguish it from the others.

The main contributions to this study are as follows:Improving histological images in an overlapping manner between an average filter and the CLAHE method.Eliminating redundant and unnecessary features produced from GoogLeNet and VGG-19 models and save essential features by the PCA method.Extracting handcrafted features by integrating DWT, LBP, FCH and GLCM methods featuresCombining the features of the GoogLeNet and VGG-19 models after and before reducing their high dimensions.Generating fusion feature vectors by integrating the features of the GoogLeNet and VGG-19 models with the handcrafted features.Developing effective systems to help physicians and pathologists diagnose histological images and support their decisions.

The rest of the paper is organized as follows: Section 2 discusses the previous systems relevant to diagnosing lung and colon cancer. Section 3 presents our proposed systems’ various tools and techniques for analyzing histological images for the LC25000 dataset. Section 4 discusses the implementation performance of all systems and summarizes their results. Section 5 discusses the performance of systems and compares algorithms with their results. Section 6 concludes the paper.

## 2. Related Work

Kwabena et al. [10] proposed DHSCapsNet to analyze histological images of lung and colon cancer. The network consists of the fusion of DHSCaps with encoder features. The features of the convolutional layers that contain strong information are grouped into the encoder features. HSquash extracts information from a variety of backgrounds. Sanidhya et al. [11] proposed a CNN Pre-Trained Diagnostic Network for Lung and Colon Cancer. Shallow CNN architecture was used to analyze histological slips. The network achieved 96% and 97% accuracy for diagnosing colon and lung cancers, respectively. Mumtaz et al. [12] proposed a capsule network with multiple inputs to construct a diagnostic model for abnormal cell carcinoma of the lung and colon. The capsule network used two masses, a convolutional layer block and a separate convolutional layer block. The convolutional layer block (CLB) takes pathological images as input, while the Separable CLB takes histopathological images for processing. SHAHID et al. [13] provided an effective model for accurately diagnosing lung and colon cancer cells. AlexNet was tuned by modifying the four essential layers and then training the dataset, which achieved an accuracy of 89%. Mesut et al. [14] proposed the DarkNet-19 model to train the lung and colon cancer dataset from scratch. The Equilibrium algorithm was applied to select the inefficient features and then sort the inefficient features from the efficient ones. Efficient features are fed to the SVM for classification. Mehedi et al. [15] proposed a deep learning model for diagnosing five lung and colon cancer classes. Images are optimized, and 2D Fourier and 2D wavelet features were applied to extract features. The model reached an accuracy of 96.33%. Ben et al. [16] applied four pre-trained deep learning models to classify areas affected by colon cancer through histological images of the AiCOLO dataset. SegNet and UNet have implemented pixel segmentation. The pre-trained ResNet achieved an accuracy of 96.98%. Md et al. [17] proposed a deep learning based on segmentation problems for abnormal cell detection to measure biomarkers for colon cancer diagnosis. The model generates annotations using a tool to help physicians diagnose conditions. Devvi et al. [18] proposed a pre-trained ResNet model for colon cancer diagnosis. The models were tested on the dataset divided into 20% and 40%, with ResNet50 getting better results than ResNet18. ResNet50 has achieved a sensitivity of 87% and an accuracy of 80%. Changjiang et al. [19] developed a system that uses labels to classify WSI images by combining the features of different magnification factors for WSI images. The network achieved an accuracy of 94.6% for grading colorectal cancer. Lin et al. [20] presented a deep learning network for segmentation of H&E-stained images of histology of the colon cancer dataset for early diagnosis. The network achieved an accuracy of 94.8% for images of tumor tissue. Dipanjan et al. [21] developed a 1D CNN network for classifying small cell lung tumors. Images were fed to extract and combine the hybrid features with the clinical features. The method achieved results that exceeded the techniques of machine learning. Vinod et al. [22] designed a method to identify pulmonary nodules from regions of interest. The watershed algorithm and Gabor filter were applied to segment the lung regions. Features were extracted and categorized by SVM. Won et al. [23] developed a DeepRePath network based on CNN to predict the stages of lung adenocarcinoma. The network was trained using the Genome Atlas dataset and validated by images of patients in the first and second stages. DeepRePath achieved an AUC of 77%. Mizuho et al. [24] presented machine learning algorithms for diagnosing three types of lung cancer. Features were extracted by homology-based image and texture analysis methods. Machine learning algorithms classify the features extracted from the two methods. Machine learning with the features of the homology-based image method is better than the features of the texture analysis method.

The researchers aimed to achieve promising results in diagnosing lung and colon cancer. Because of the similar characteristics of the types of tumors in their early stages, promising accuracy is still the goal of all researchers. Therefore, this study focused on extracting features from deep learning models and integrating them to represent each class with representative features. It also combined the features of deep learning models with handcrafted features to achieve promising results.

## 3. Materials and Methods

This section discusses the tools and techniques applied in this work to diagnose histological images of the LC25000 dataset (Figure 1). All images were subjected to an averaging filter and the CLAHE method to remove noise and increase the contrast of low-light areas. The PCA method was applied after each GoogLeNet and VGG-19 model to remove redundant and unimportant features and save essential features. The first strategy used ANN with the important features of GoogLeNet and VGG-19. The second strategy used ANN with hybrid features of GoogLeNet and VGG-19, where features are combined before and after PCA. The third strategy used the fusion features of the CNN models (GoogLeNet and VGG-19) and the handcrafted features.

### 3.1. Description of the LC25000 Dataset

This study collected a dataset of histopathological images of lung and colon cancer called LC25000 from the publicly available Kaggle website to evaluate the proposed systems [25]. The dataset was compiled by Andrew Borkowski and his associates at James Hospital Tampa, Florida, which consists of 25,000 images divided into two types of colon cancer and three types of lung cancer. The images are distributed among the five types equally, meaning the dataset is balanced, and each type contains 5000 images. These types are colon_aca (Adenocarcinoma) and colon_bnt (Benign Tissue), lung_aca (Adenocarcinoma), lung_bnt (Benign Tissue), and lung_scc (Squamous Cell Carcinoma). Colon adenocarcinoma accounts for more than 95% of colon cancers due to the non-removal of polyps in the large intestine. Adenocarcinoma of the lung accounts for more than 40% of lung cancers, which appear in glandular cells and spread within the lung and alveoli. Lung squamous cell carcinoma accounts for more than 30% and is the second most common type of lung cancer, which appears in the bronchi. The other two types are benign and do not spread to other parts of the body. However, it must be effectively verified by biopsy and removal. Figure 2a shows the samples of the LC25000 dataset for the five classes.

### 3.2. Improving Histological Images for the LC25000 Dataset

Preprocessing is necessary to improve images by removing noise and optimizing important properties to extract critical information from images and to make them compatible with deep learning and machine networks. Images of the LC25000 dataset contain noise due to biopsy mixed with various medical materials and poor contrast between affected and surrounding tissues. Therefore, all images of the LC25000 dataset in this study were subjected to consecutive treatments. RGB is adjusted by averaging each channel [26]. The color consistency was also adjusted by scaling the color consistency of the texture of each image. An average filter was applied to each histological image of lung and colon cancers to improve their appearance. Each time the filter takes 16 pixels (the center pixel and its neighbors), it calculates averages of 15 pixels and replaces them with the center pixel. The process is repeated, and the filter moves to select another 16 pixels, and so the process continues until each pixel is targeted in the image, as in Equation (1).
(1)$$z\left(i\right)=\frac{1}{N}\overset{N-1}{\underset{j=0}{\sum }}x\left(i-j\right)\;$$
where $z\left(i\right)$ refers to inputted, $x\left(i-j\right)$ refers to the first input, and *N* refers to the number of pixels.

After the removal of artifacts, tissue contrast is increased by CLAHE technology. The technology improves the non-luminous pixels by showing bright pixels and distributing them over dark areas. The technology targets a central pixel with a certain number of neighbors and compares it with its neighbors. When the central pixel’s value is greater than its neighbors, the technique increases the contrast. On the other hand, when the value of the central pixel is smaller than its neighbors, the method reduces the contrast [27]. The process is repeated and moves to select another pixel with its neighbors, and thus the process continues until every pixel in the image is targeted. Figure 2b describes histological samples after improvement.

### 3.3. ANN Network with CNN Features

This section describes the development of a hybrid technique for histological image diagnostics for the LC25000 dataset in its early stages. Because deep learning models are late in training datasets on expensive computers, this technique solves this challenge. The main idea of this technique is to extract features from CNN models and feed them to PCA to select essential features and remove unimportant and redundant features [28]. Essential features are classified by ANN very efficiently.

#### 3.3.1. CNN Features 

In recent years, biomedical images have received widespread interest from CNNs, which consist of feature extraction layers and layers for classification. The greater the depth of CNN networks, the better the performance; however, the increase in depth leads to fading resolution and color, which is a challenge for CNNs. CNN models are superior to machine learning algorithms by having dozens of convolutional layers for feature extraction with high accuracy, pooling layers to reduce high dimensionality, and some auxiliary layers to do a specific task. This study fed the improved images to GoogLeNet and VGG-19 models. The two models analyze the image data of the LC25000 dataset with great accuracy and distinguish each of the five classes with unique features [29]. Each layer in the model has a task assigned to it to extract features. Each layer has millions of neurons interconnected with specific parameters.

Pooling layers reduce the high dimensions produced by convolutional layers through two methods: max-pooling and average pooling. The max-pooling layers reduce the high dimensions by representing a group of pixels by their max value, as in Equation (2). In contrast, the average pooling layers reduce the high dimensions by representing a group of pixels with their average, as in Equation (3) [30].
(2)$$z\left(i;\;j\right)=max_{m,n=1\ldots .k}\;f\left[\left(i-1\right)p+m;\;\left(\;j-1\right)p+n\right]$$
(3)$$z\left(i;\;j\right)=\frac{1}{k^{2}}\underset{m,n=1\ldots .k}{\sum }f\left[\left(i-1\right)p+m;\;\left(\;j-1\right)p+n\right]$$
where m, n refer to the placement in the matrix, *p* refers to the step of filter, *f* refers to the size of the filter, and *k* refers to the features vectors.

Auxiliary layers, such as ReLU layers, follow convolutional layers for further improvement. The layers suppress negative values, convert them to zero, and pass positive values.

CNN layers produce millions of parameters and connections, causing overfitting; however, this challenge is addressed by a dropout layer that passes 50% of the neurons at a time. On the other hand, this layer increases the training rate to double.

#### 3.3.2. The ANN Network

The high-dimensional features were extracted from the GoogLeNet and VGG-19 models and fed to PCA to select the essential features, delete the redundant and non-significant features, and save them in vectors of size 25,000 × 455 for the GoogLeNet and 25,000 × 455 for the VGG-19. These vectors are sent to the ANN. The ANN receives the GoogLeNet and VGG-19 features separately and performs a classification task to classify the input features into five classes. The ANN input layer consists of 455 input units based on the number of features. ANN contains 20 hidden layers connected to parameters (weights) at the layer level and other layers in which the required tasks are solved. The process is repeated from the first hidden layer to the last; the weights are adjusted, and the minimum square error (MSE) is calculated based on the difference between the actual $x_{i}$ and expected $y_{i}$ values, as in Equation (4) [31]. The output layer sorts the features of each image with its appropriate class, producing five neurons, each representing a class of dataset.
(4)$$\;\mathrm{MS}E=\frac{1}{m}\overset{m}{\underset{i=1}{\sum }}\;{\left(\;x_{i}-y_{i}\right)}^{2}$$
where m refers to the number of features, $x_{i}$ refers to the actual value, and $y_{i}$ refers to the expected value.

Figure 3 shows the basic structure of the histological diagnostic methodology for the LC25000 dataset and the discrimination of early-stage tumor types by ANN with the features of the GoogLeNet and VGG-19 models.

### 3.4. ANN with Fusion Features of CNN

This section develops a hybrid technique for early diagnosis of histological images of the LC25000 dataset and early discrimination of its types. Since deep learning models are slow to train datasets on expensive computers, this technique solves this challenge. This technique has two systems; the main idea of the first system is to extract the features from the CNN models (GoogLeNet and VGG-19) and send them to PCA separately to select the important features, remove the non-significant and redundant features and then merge the features of the GoogLeNet and VGG-19; this is called feature merge after PCA [32]. The main idea of the second system is to extract the features from the CNN models (GoogLeNet and VGG-19) and then combine them and send them to PCA to determine the essential features and remove the unimportant and redundant features; this is called feature merge before PCA [33]. The basic features are fed to the ANN to split into training and validation samples and test their performance on the test dataset.

Figure 4 illustrates the two approaches for diagnosing lung and colon cancer histological images and early discrimination among their types.

The last layers of GoogLeNet and VGG-19 produce high-level features for two models: (7, 7, 512) and (7, 7, 512). The Global Average Pooling layer converts an image from high-level features into a distinctive feature of 4096 features of both models.

The steps of implementing the first approach: First, increasing the contrast of the affected histological image areas and eliminating noise using enhancement filters. Second, the improved images are passed to GoogLeNet and VGG-19 models to analyze images by convolutional layers and pooling for spatial feature extraction and saving in a size of 25,000 × 4096 for the GoogLeNet and 25,000 × 4069 for the VGG-19. Third, applying PCA for selecting critical features, removing non-important and duplicate features of GoogLeNet and saving them in vectors of size 25,000 × 455. Fourth, applying PCA for selecting critical features, removing non-important and duplicate features of VGG-19, and saving them in vectors of size 25,000 × 455. Fifth, integrating critical features of low dimensions for GoogLeNet and VGG-19 and saving them in vectors of size 25,000 × 910. Sixth, the ANN receives feature vectors of 25,000 × 910 and divides them into training, validation, and performance testing samples.

The steps for implementing the second approach are as follows: The first and second steps are as in the first approach. Third, combining the high-dimensional features of GoogLeNet and VGG-19 models and saving them in vectors of size 25,000 × 4096. Fourth, applying PCA to select critical features, remove non-important and redundant features, and then saving them in vectors of size 25,000 × 740. Fifth, the ANN receives feature vectors of size 25,000 × 740 and then divides them into training, validation, and performance testing samples.

### 3.5. ANN with Fusion Features of CNN and Handcrafted

In this section, a hybrid technique is developed for the early diagnosis of histological images of the LC25000 dataset and early discrimination of its types. This technique has two systems; the main idea is fusion feature extraction and classification by ANN [34]. Fusion features are fusion features of CNN models with handcrafted features. The handcrafted features are fusion features extracted from the DWT, LBP, FCH and GLCM methods. The technology consists of two systems, as shown in Figure 5: the first system extracts features from CNN models (GoogLeNet and VGG-19) and sends them to PCA separately to select essential features and remove non-important and redundant features, then combines GoogLeNet features with handcrafted features and combines the features of the VGG-19 with the handcrafted features. The fusion features are fed to the ANN for splitting into training and validation samples and testing their performance on the test dataset.

The steps of achievement of the approaches: First, increasing the contrast of the affected histological image region and removing noise using improvement filters. Second, sending the improved images to GoogLeNet and VGG-19 for analysis by convolutional layers for spatial feature extraction and saving in the size of 25,000 × 4096 for both the GoogLeNet and VGG-19 separately. Third, using the PCA method for selecting essential features, removing non-important and repeated features from GoogLeNet and VGG-19 separately and then saving them in vectors with sizes of 25,000 × 455 for GoogLeNet and 25,000 × 455 for VGG-19.

Fourth, extracting color, texture, and geometric features by DWT, LBP, FCH, and GLCM methods is as follows.

The DWT method receives ROI images of the LC25000 dataset for analysis and extraction of the most important geometric features. The mechanism of the method works with four filters; each filter works to analyze a specific part of the image. Thus, the image is divided into four parts, and a specific filter is passed to each part. The low filter analyzes the components of the first part of the image and extracts approximate parameters through three measures of variance, mean, and standard deviation. Low-High and High-Low filters analyze the second and third components of the image and extract detailed parameters through three measures of variance, mean, and standard deviation for each part of the image. The high filter analyzes the components of the fourth part of the image and extracts detailed parameters through three measures of variance, mean, and standard deviation. Thus, the total feature output of method is 12 features, saved in vectors of size 25,000 × 12.

The LBP method receives the ROI of the LC25000 dataset images to convert them to grayscale images, analyze them, and extract the most critical features of the binary surface texture. The algorithm extracts spatial information of histological images of lung and breast cancer. In this work, the algorithm is set to 4 × 4, which means that in each iteration, a central pixel $g_{c}$ is replaced by 15 adjacent pixels $g_{p}$ according to Equation (5) [34]. The algorithm continues until each pixel in the image is replaced. Thus, the total feature output of the method is 203 features saved in vectors of size 25,000 × 203.
(5)$$LBP_{R,P}=\overset{P-1}{\underset{p=0}{\sum }}s\left(g_{p}-g_{c}\right)2^{p}$$
where $g_{c}$ refers to the aim pixel, R to the closest radius, $g_{p}$ to the closest pixels and *P* to the number of adjoining pixels.

The FCH method receives ROI images of the LC25000 dataset to analyze and extract the essential color features. The algorithm operates by the fuzzy logic method. The characteristics of the color are essential to distinguish each type of tumor [35]. The method creates many histogram bins and assigns each bin histogram to a specific color. The colors of each image are represented in the bins histogram; any two colors in the same bin histogram are the same. Thus, the total output of method 16 is distinct and is saved in vectors of size 25,000 × 16.

The GLCM method receives the ROI of the LC25000 dataset images to convert them to grayscale images, analyze them, and extract the most critical rough and smooth texture features. The algorithm extracts spatial information by comparing the pixel and its neighbors. The central pixel and its neighbors are analyzed based on distance d and angle 0°, 45°, 90°, 135° [36]. The coarse region has different pixel values, while the smooth regions have equal pixel values. Thus, the total feature product of the method is 13 features, which are saved in vectors of size 25,000 × 13.

Fifth, the features of all the DWT, LBP, FCH, and GLCM methods are combined to produce the handcrafted features and stored on 25,000 × 244 vectors.

Sixth, the essential features of the GoogLeNet are combined with the handcrafted features to produce the fusion features and save them in vectors of size 25,000 × 699.

Seventh, the essential features of the VGG-19 are combined with the handcrafted features to produce the fusion features and preserve them in vectors of size 25,000 × 699.

Eighth, the ANN receives fusion feature vectors of 25,000 × 699 for both systems and divides them into training, validation, and performance testing samples.

## 4. The Results of the System Execution

### 4.1. Split of LC25000 Dataset

This study applied the systems to histological images of the LC25000 dataset for diagnosing lung and colon cancer and their discrimination in its early stages. The LC25000 dataset contains 25,000 histological images taken from the patient’s affected tissue by biopsy. The dataset is divided into five types of malignant and benign tumors of lung and colon cancer, distributed as follows: 5000 histological images of Colon Adenocarcinoma, 5000 histological images of Colon Benign Tissue, 5000 histological images of Lung Adenocarcinoma, 5000 histological images of Lung Benign Tissue and 5000 histological images of Lung Squamous Cell Carcinoma. Thus, it is noted that the dataset contains three types of malignant tumors and two types of benign tumors. All dataset classes are balanced and have the same number of histological images. The dataset was divided into 80% during systems training and validation (80:20), and 20% of the dataset was kept as a test dataset to evaluate the performance of the systems, as shown in Table 1.

### 4.2. Evaluation Metrics

All systems for histological diagnosis of lung and colon cancer were evaluated with the same measures of sensitivity, precision, accuracy, specificity, and AUC indicated by Equations (6)–(10). It is noted that the equations contain variables such as TP and TN, which mean the number of samples classified correctly, and FP and FN, which mean the number of samples classified incorrectly [37]. All of these variables are obtained from the confusion matrix, which is produced as an output to evaluate the performance of each system.
(6)$$\mathrm{Sensitivity}=\frac{\mathrm{TP}}{\mathrm{TP}+\mathrm{FN}}\times 100\%$$
(7)$$\mathrm{Precision}=\frac{\mathrm{TP}}{\mathrm{TP}+\mathrm{FP}}\times 100\%$$
(8)$$\mathrm{Accuracy}=\frac{\mathrm{TN}+\mathrm{TP}}{\mathrm{TN}+\mathrm{TP}+\mathrm{FN}+\mathrm{FP}}\times 100\%$$
(9)$$\mathrm{Specificity}=\frac{\mathrm{TN}}{\mathrm{TN}+\mathrm{FP}}\times 100$$
(10)$$\mathrm{AUC}\;=\frac{\mathrm{TP}\;\mathrm{Rate}}{\mathrm{FP}\;\mathrm{Rate}}$$

### 4.3. Results of ANN with CNN Features 

The section discusses the summary of results achieved by ANN when fed with GoogLeNet and VGG-19 features after removing non-significant and redundant features to reduce the high-dimensionality of histological images of the lung and colon cancer dataset. ANN receives critical features, trains them, adjusts weights during validation, and evaluates their performance on test datasets to achieve promising results for early lung and colon cancer detection and discrimination.

Table 2 and Figure 6 summarize the performance of ANN execution when fed with significant low-dimensionality features of GoogLeNet and VGG-19 with a size of 25,000 × 455 separately for each model. When classifying the important low-dimensional features of the GoogLeNet model by ANN, it achieved a sensitivity of 96.49%, a precision of 96.19%, an accuracy of 95.50%, a specificity of 99.12%, and an AUC of 97.67%. The ANN network, when classifying the important low-dimensional features of the VGG-19 model, achieved a sensitivity of 95.88%, a precision of 96.18%, an accuracy of 95.92%, a specificity of 98.48%, and an AUC of 98.02%.

Figure 7 summarizes the confusion matrix resulting from the implementation of the ANN when classifying the significant low-dimensional features of GoogLeNet and VGG-19 at a size of 25,000 × 455 separately for each model. When classifying important low-dimensional features of the GoogLeNet model by ANN, it reached an accuracy for each type of tumor as follows: colon_aca class of 95.1%, colon_bnt class of 94.2%, lung_aca class of 97.3%, lung_bnt class of 94.8%, and lung_scc class of 96.1%. In contrast, when ANN classified the significant low-dimensional features of the VGG-19 model, the accuracy for each tumor type was achieved as follows: colon_aca class accuracy of 97.1%, colon_bnt class of 90.8%, lung_aca class of 97.9%, and lung_bnt class of 96.2% the lung_scc class of 97.6%.

### 4.4. Results of ANN with Fusion Features of CNN

The section discusses a summary of the results achieved by ANN when fed with combined features from the GoogLeNet and VGG-19 models. Two systems were implemented based on feature merging. The first system was based on extracting the features of two models and reducing their dimensions separately by PCA, then merging the low-dimensional features. The second system was based on extracting features from the two models, combining high-dimensional features, and removing unimportant and redundant features to reduce high-dimensionality by PCA. The ANN receives merged important features, trains them, adjusts weights during validation, and evaluates its performance on test datasets to achieve promising results for early detection and discrimination of lung and colon cancer.

Table 3 and Figure 8 summarize ANN execution performance when fed with mixed features after PCA for GoogLeNet and VGG-19 with a size of 25,000 × 740 and mixed features before PCA for GoogLeNet and VGG-19 with a size of 25,000 × 740. When classifying mixed features after PCA of GoogLeNet and VGG-19 by ANN, it achieved a sensitivity of 98.75%, a precision of 98.76%, an accuracy of 98.66%, a specificity of 99.60%, and an AUC of 99.58%. In contrast, when classifying mixed features before PCA of GoogLeNet and VGG -19, ANN achieved a sensitivity of 98.41%, a precision of 98.81%, an accuracy of 98.54%, a specificity of 99.64%, and an AUC of 99.45%.

Figure 9 summarizes the confusion matrix generated by the implementation of the ANN when classifying the low-dimensional mixture features of GoogLeNet and VGG-19. When classifying the mixed features after PCA of GoogLeNet and VGG-19 by ANN, an accuracy was reached for each tumor type as follows: Colon_aca class of 100%, colon_bent class of 99.5%, Lung_aca class of 96.8%, Lung_bnt class of 100%, and Lung_scc class of 97%. In contrast, when classifying the mixed features before PCA of GoogLeNet and VGG-19 by ANN, an accuracy for each tumor type was reached as follows: colon class of 99.1%, colon_bent class of 98.9%, Lung_aca class of 98.3%, Lung_bnt class of 98.3%, and Lung_scc class of 98.1%.

### 4.5. Results of ANN with Fusion Features of CNN and Handcrafted

The section discusses a summary of results achieved by ANN when fed with fusion features combined from GoogLeNet and handcrafted features, fusion features combined from VGG-19 and handcrafted features. Two systems were implemented based on feature merging. The first system was based on extracting GoogLeNet features, reducing their dimensions and combining them with handcrafted features. The second system was based on extracting VGG-19 features, reducing their dimensions and combining them with handcrafted features. The ANN receives fusion task features, trains them, adjusts weights during validation, and evaluates its performance on test datasets to achieve promising results for early detection and discrimination of lung and colon cancer.

Table 4 and Figure 10 summarize the performance of the ANN implementation when fed with fusion features created from features of GoogLeNet and handcrafted features with a size of 25,000 × 699, and fusion features created from features of VGG-19 and handcrafted features with a size of 25,000 × 699. When categorizing the fusion features of GoogLeNet and handcrafted features by ANN, they reached a sensitivity of 99.42%, a precision of 99.54%, an accuracy of 99.22%, a specificity of 99.85%, and an AUC of 99.74%. In contrast, when classifying the fusion features of VGG-19 and handcrafted features, the ANN achieved a sensitivity of 99.85%, a precision of 100%, an accuracy of 99.64%, a specificity of 100%, and an AUC of 99.86%.

Figure 11 summarizes the confusion matrix generated by implementing the ANN when classifying low-dimensional fusion features. When classifying the fusion features of GoogLeNet and handcrafted features using ANN, an accuracy was reached for each tumor type: Colon_aca class of 99%, colon_bent class of 98.9%, Lung_aca class of 99.6%, Lung_bnt class of 99%, and Lung_scc class of 99.6%. In contrast, when classifying the fusion features of VGG-19 and handcrafted features using ANN, an accuracy was reached for each tumor type as follows: Colon_aca class of 99.7%, colon_bent class of 99.4%, Lung_aca class of 99.6%, Lung_bnt class of 99.7%, and Lung_scc class of 99.8%.

There are some standard tools for evaluating an ANN as follows:

#### 4.5.1. Receiver Operating Characteristic (ROC)

The ROC or AUC is one of the standard tools of the ANN to evaluate its performance on the LC25000 dataset for early diagnosis of lung and colon cancer. This tool assesses the ANN by analyzing the LC25000 dataset and calculating both TP and FN rates. Figure 12 shows the performance of the ANN in analyzing the LC25000 dataset by dividing the *y*-axis TP rates by the *x*-axis FP rates. The grid performs best when the AUC approaches one or approaches a left angle aligned with the *y*-axis. When classifying the fusion features of GoogLeNet and handcrafted features using ANN, it achieved an AUC of 99.74%. In contrast, when classifying the fusion features of VGG-19 and handcrafted features using ANN, it achieved an AUC of 99.86%.

#### 4.5.2. Cross-Entropy

Cross-entropy is one of the standard tools of ANN to evaluate its performance on the LC25000 dataset for early diagnosis of lung and colon cancer. This tool evaluates the network through the dataset’s images by calculating the least error between the expected and actual values. The lowest error is recorded in each epoch, and the network continues until all epochs are completed. Then the lowest error obtained by the network during any epoch is sorted, and the best performance achieved during the epoch is recorded [38]. Figure 13 shows the performance of the ANN in analyzing the LC25000 dataset by cross-entropy. The figure shows different colors, each representing the implementation of the ANN for a particular stage of the LC25000 dataset. The training dataset is in blue, and the validation dataset is in green. The test dataset was reserved to evaluate the network performance and is represented by the red color. When classifying the fusion features of GoogLeNet and handcrafted features using ANN, the best validation at 0.012885 was reached at epoch 29. In contrast, when classifying the fusion features of VGG-19 and handcrafted features using ANN, a validation best at 0.013046 was reached at epoch era 32.

#### 4.5.3. Error Histogram

The error histogram is one of the standard tools of the ANN to evaluate its performance on the LC25000 dataset for early diagnosis of lung and colon cancer. This tool evaluates the ANN by analyzing the LC25000 dataset and calculating the target and output values error. In each iteration, the error is computed according to the instances of the dataset [39]. Figure 14 shows the performance of the ANN for analyzing the LC25000 dataset by error histogram. The figure shows different colors, each representing the implementation of the ANN for a particular stage of the LC25000 dataset. The training dataset is in blue, and the validation dataset is in green. The test dataset was reserved to evaluate the network performance and is represented in red. When classifying the fusion features of GoogLeNet and handcrafted features using ANN, the best performance is obtained between the values 0.9325 and −0.9321 among 20 bins. In contrast, when classifying the fusion features of VGG-19 and handcrafted features using ANN, the best performance was obtained between values of 0.9456 and −0.9447 among 20 bins.

#### 4.5.4. Validation Checks and Gradient 

Validation and gradient are the standard tools of ANN to evaluate its performance on the LC25000 dataset for early diagnosis of lung and colon cancer. This tool assesses the ANN by analyzing the LC25000 dataset by calculating the gradient and values that fail through each epoch to see the best check during any epoch [40]. Figure 15 shows the performance of the ANN for analyzing the LC25000 dataset through validation and gradient. When classifying the fusion features of GoogLeNet and handcrafted features using ANN, the best performance was obtained in a gradient of 0.0030438 at epoch 35; the best validation is 6. In contrast, when classifying the fusion features of VGG-19 and handcrafted features using ANN, the best performance was obtained in a gradient of 0.0015367 at epoch 38; the best validation is 6.

#### 4.5.5. Regression

Regression is one of the standard tools of ANN to evaluate its performance on the LC25000 dataset for early diagnosis of lung and colon cancer. This tool assesses the ANN by analyzing the LC25000 dataset by calculating continuous variables according to other variables [41]. Figure 14 shows the performance of the ANN for analyzing the LC25000 dataset by predicting the *x*-axis-represented target values according to the *y*-axis-represented output values. The network performs best when the gradient is close to 1. When classifying the fusion features of GoogLeNet and handcrafted features using ANN, they reach a regression rate of 97.97% during the training of the dataset, 97.38% during its validation, and 97.51% during its performance testing [42]. In contrast, when classifying the fusion features of VGG-19 and handcrafted features using ANN, it reaches a regression rate of 96.90% during the training of the dataset, 96.95% during its validation, and 96.75% during its performance testing as shown in Figure 16.

## 5. Discussion of the Performance of the Systems

Lung and colon tumors are among the most common types that must be detected early. This study discussed many effective systems that can detect lung and colon tumors early and distinguish between them. Tumors are similar in their early stages, which is an obstacle for doctors to distinguish the type of tumor, and therefore artificial intelligence techniques came to solve this challenge. Because of the similarity of early-stage tumors, this study focuses on extracting features in several ways and integrating them. Three strategies have been developed, each with two systems, with different methods and algorithms, aiming to achieve superior accuracy for early diagnosis and discrimination of tumor types in the LC25000 dataset.

Histological images of the LC25000 dataset were improved using an averaging filter and the CLAHE method. The first strategy has two systems for diagnosing lung and colon cancer and early discrimination between the types of tumor. The improved images of GoogLeNet and VGG-19 were fed for analysis by several convolutional layers and saved in vectors. Vectors contain redundant and unimportant features, so PCA was applied to reduce high dimensions by eliminating them and saving them in vectors with a size of 25,000 × 455 for each model. These vectors were submitted to the ANN Network to classify them into five classes. With the selected features of GoogLeNet, ANN achieved 95.5% accuracy. In contrast, ANN achieved 95.92% accuracy with the selected features of VGG-19.

The second strategy has two systems for diagnosing lung and colon cancer and early discrimination between the types of tumors. The optimized images of GoogLeNet and VGG-19 were fed for analysis by several convolutional layers and saved in vectors. After extracting the high-dimensional features of the two models, there are two ways to combine the features of the two models. The first method is to reduce high dimensionality by eliminating unimportant and redundant features by PCA for each model separately. Then the selected and important features of the two models are combined and saved in vectors of size 25,000 × 910. The second method combines the high-dimensional features of GoogLeNet and VGG-19 models in vectors of size 25,000 × 4096. Then the redundant and unimportant features are eliminated using PCA and saved in vectors of a size of 25,000 × 740. When feeding the ANN with feature vectors with a size of 25,000 × 910, it achieved an accuracy of 98.7%. When feeding the ANN with feature vectors with a size of 25,000 × 740, it achieved an accuracy of 98.5%.

The third strategy has two systems for diagnosing lung and colon cancer and early discrimination of tumor types. Images optimized for GoogLeNet and VGG-19 were fed for analysis by several convolutional layers and saved in vectors. Vectors have redundant and non-significant features, so PCA was implemented to reduce high dimensionality by eliminating them and saving them to a vectors of a size of 25,000 × 455 for each model. The features of the DWT, LBP, FCH and GLCM methods are combined into vectors of 25,000 × 244, called handcrafted features. The selected and essential features of the model (GoogLeNet and VGG-19) were combined with the handmade features called fusion features. With the fusion features of the GoogLeNet and handcrafted features, the ANN achieved an accuracy of 99.22%. In contrast, with the fusion features of the VGG-19 and the handcrafted features, the ANN achieved an accuracy of 99.64%.

Table 5 and Figure 17 summarize the performance of ANN implementation with combined features from CNN and traditional methods for classifying histological images for the LC25000 dataset. The table contains the accuracy of each system and the accuracy achieved by each system for diagnosing each class. All systems produced promising results, where ANN with fusion features of the GoogLeNet model and handcrafted features attained an accuracy of 99.64%. The best accuracy for each type (class) for the LC25000 dataset is as follows: for classes colon_aca, colon_bnt, lung_aca, and lung_scc, f 99.7%, 99.4%, 99.6% and 99.8%, respectively, by ANN with fusion features of GoogLeNet and handcrafted features. The best accuracy for the lung_bnt class is 100% by ANN with the combination of GoogLeNet and VGG-19 features.

Table 6 shows the performance results of the current systems related to the classification of the LC25000 dataset and compares it with the performance of the proposed system. It is noted that the performance of the proposed system is superior to all systems of previous studies.

## 6. Conclusions

Lung and colon cancer are the most common types and lead to death. There is a chance of survival if it is caught early. This study aims to detect lung and colon cancer early by developing three strategies, each of which has two systems. The first strategy is to diagnose the LC25000 dataset by ANN with the features of the GoogLeNet and VGG-19 models after deleting the repetitive and non-significant features of dimensionality reduction by PCA. The second strategy is LC25000 dataset diagnosis by ANN with mixed features of GoogLeNet and VGG-19. The third strategy is to diagnose the LC25000 dataset by ANN with fusion features of CNN models and handcrafted features. The proposed systems achieved superior performance for early diagnosis of LC25000 dataset images. With the fusion features of the VGG-19 and handcrafted features, the ANN attained a sensitivity of 99.85%, a precision of 100%, an accuracy of 99.64%, a specificity of 100%, and an AUC of 99.86%.

## Figures and Tables

**Figure 1 bioengineering-10-00383-f001:**
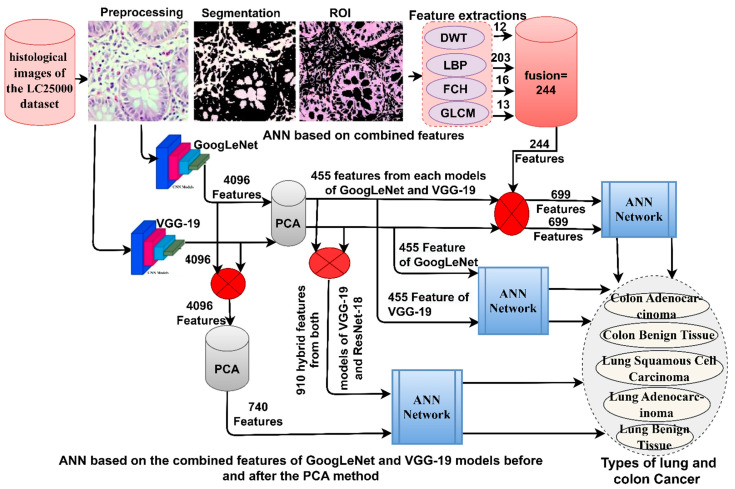
Methodologies framework for proposed systems for histological image diagnostics for the LC25000 dataset.

**Figure 2 bioengineering-10-00383-f002:**
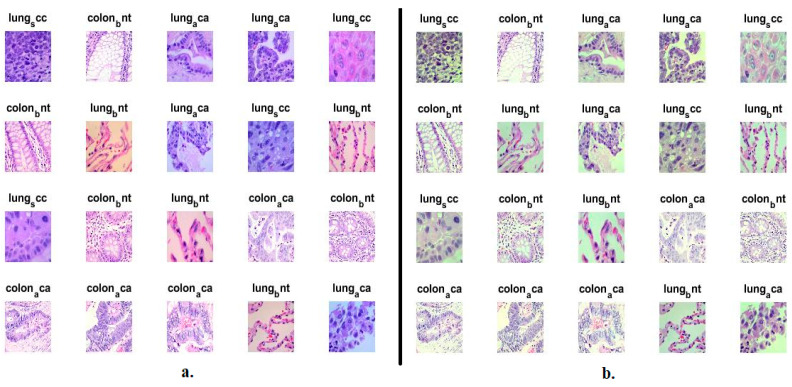
Number of samples of histological images of the LC25000 dataset (**a**) before improvement, and (**b**) after improvement.

**Figure 3 bioengineering-10-00383-f003:**
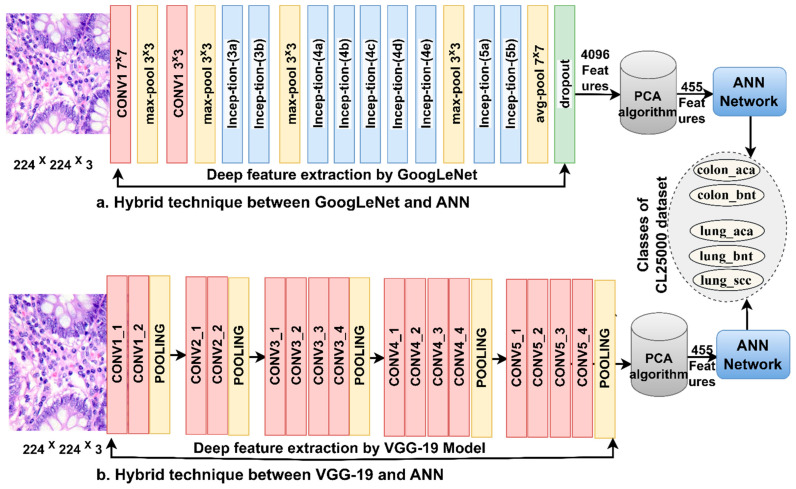
Basic methodology for histological images diagnosis of lung and colon cancer dataset by ANN with CNN features.

**Figure 4 bioengineering-10-00383-f004:**
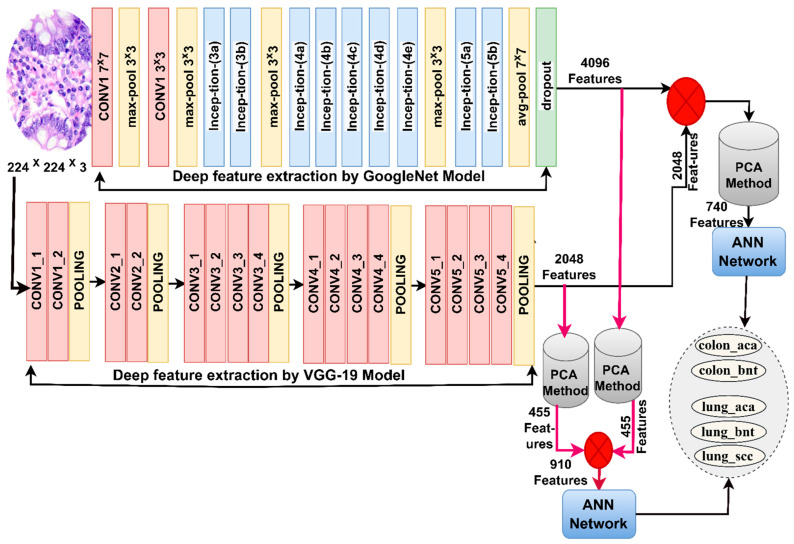
Methodology for histological image diagnostics of the LC25000 dataset by ANN with CNN merged features.

**Figure 5 bioengineering-10-00383-f005:**
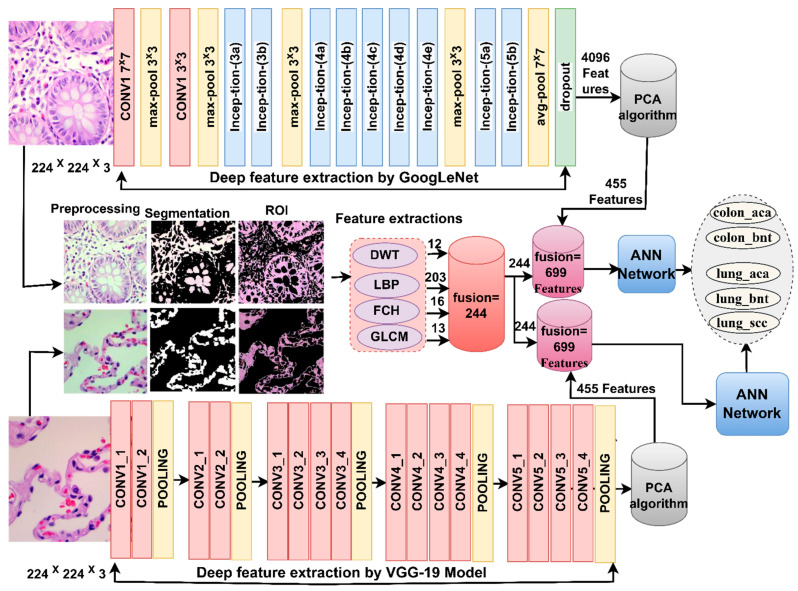
Methodology for histological image diagnostics of the LC25000 dataset by ANN with fusion features.

**Figure 6 bioengineering-10-00383-f006:**
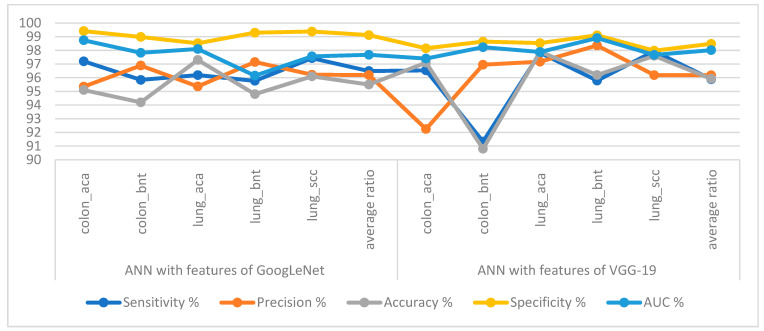
Display performance of ANN implementation with low-dimensional features of GoogLeNet and VGG-19 for histological image diagnosis of the LC25000 dataset.

**Figure 7 bioengineering-10-00383-f007:**
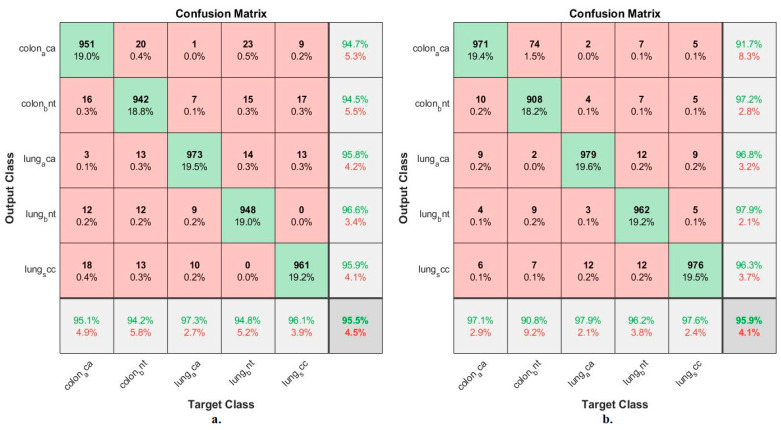
Confusion matrix for classifying histological images of the LC25000 dataset by ANN with important features of models (**a**) GoogLeNet and (**b**) VGG-19.

**Figure 8 bioengineering-10-00383-f008:**
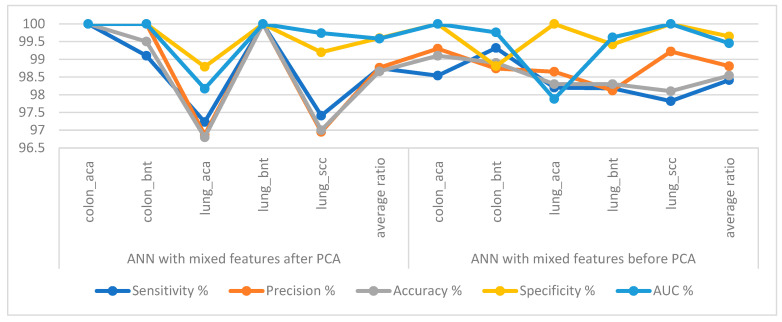
Display performance of ANN implementation with mixed features after and before applying PCA for histological image diagnosis of the LC25000 dataset.

**Figure 9 bioengineering-10-00383-f009:**
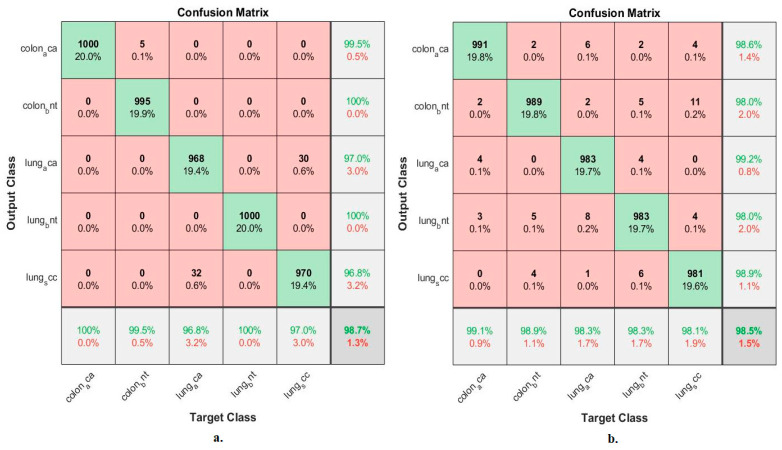
Confusion matrix for classifying histological images of the LC25000 dataset by ANN with mixed features of GoogLeNet and VGG-19 models. (**a**) Combined features after PCA. (**b**) Combined features before PCA.

**Figure 10 bioengineering-10-00383-f010:**
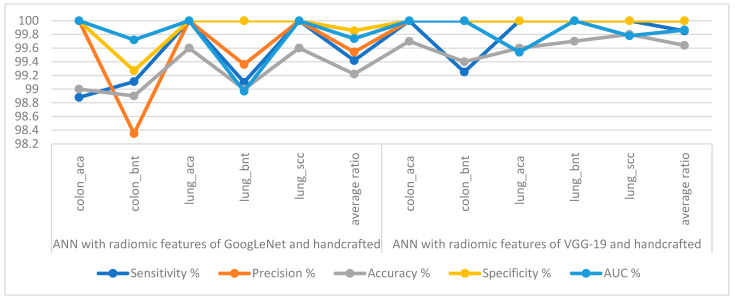
Display performance of ANN implementation with fusion features for histological image diagnosis of the LC25000 dataset.

**Figure 11 bioengineering-10-00383-f011:**
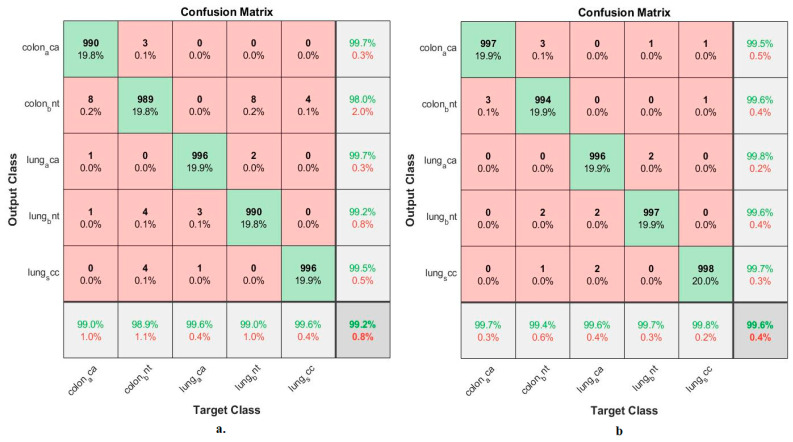
Confusion matrix to display a classification of histological images of the LC25000 dataset by ANN with fusion features of (**a**) GoogLeNet and handcrafted features, and (**b**) VGG-19 and handcrafted features.

**Figure 12 bioengineering-10-00383-f012:**
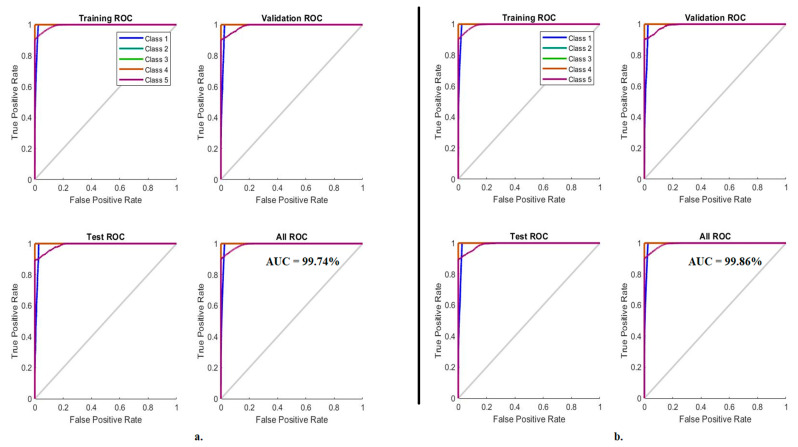
ROC to display a classification of histological images of the LC25000 dataset by ANN with fusion features of (**a**) GoogLeNet and handcrafted features, and (**b**) VGG-19 and handcrafted features.

**Figure 13 bioengineering-10-00383-f013:**
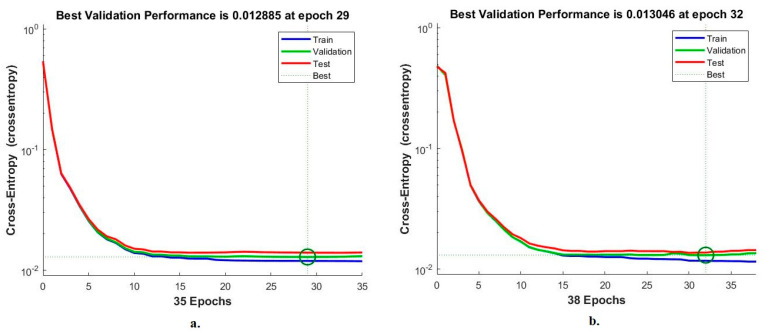
Cross-entropy to display a classification of histological images of the LC25000 dataset by ANN with fusion features of (**a**) GoogLeNet and handcrafted features, and (**b**) VGG-19 and handcrafted features.

**Figure 14 bioengineering-10-00383-f014:**
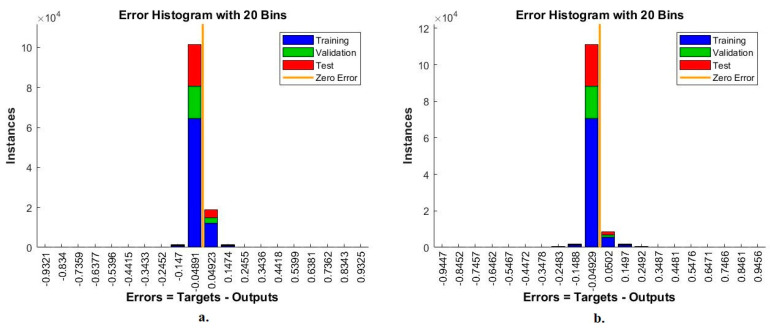
Error Histogram to display a classification of histological images of the LC25000 dataset by ANN with fusion features of (**a**) GoogLeNet and handcrafted features, and (**b**) VGG-19 and handcrafted features.

**Figure 15 bioengineering-10-00383-f015:**
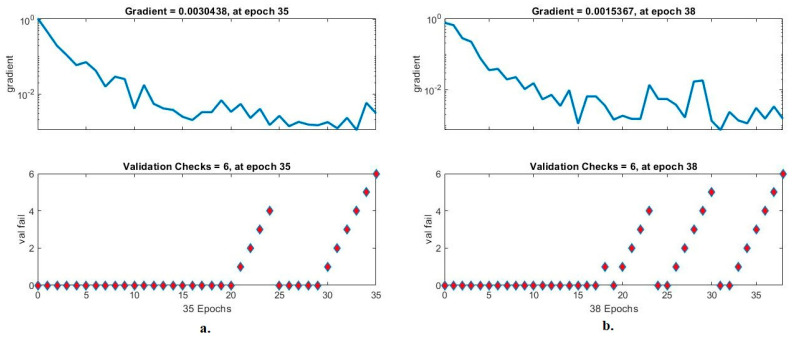
Validation checks and gradient to display a classification of histological images of the LC25000 dataset by ANN with fusion features of (**a**) GoogLeNet and handcrafted features, and (**b**) VGG-19 and handcrafted features.

**Figure 16 bioengineering-10-00383-f016:**
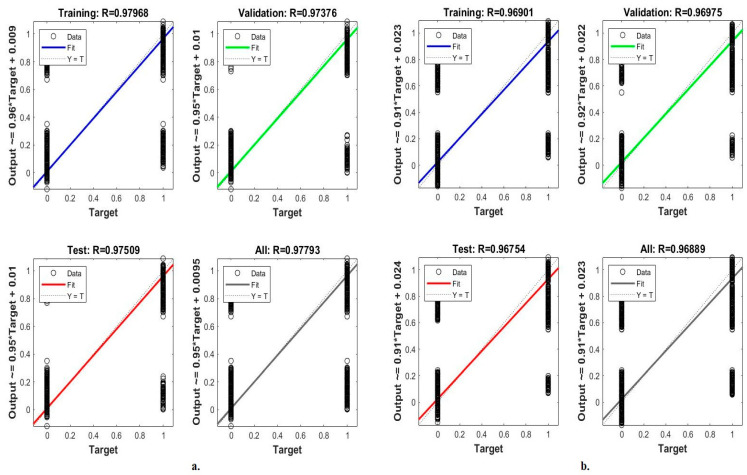
Regression to display a classification of histological images of the LC25000 dataset by ANN with fusion features of (**a**) GoogLeNet and handcrafted features, and (**b**) VGG-19 and handcrafted features.

**Figure 17 bioengineering-10-00383-f017:**
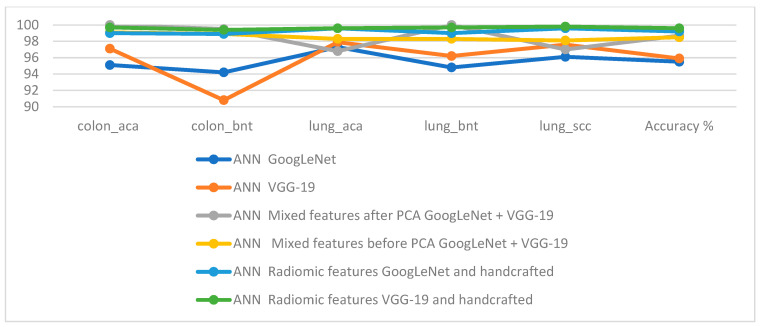
Display of the performance of the systems implemented in this work for histopathological images diagnosis of the LC25000 dataset.

**Table 1 bioengineering-10-00383-t001:** Splitting the LC25000 dataset through all phases.

Phase	(80:20) Training and Validation		Testing 20%
Classes	Training (80%)	Validation (20%)	
colon_aca	3800	800	1000
colon_bnt	3800	800	1000
lung_aca	3800	800	1000
lung_bnt	3800	800	1000
lung_scc	3800	800	1000

**Table 2 bioengineering-10-00383-t002:** Results of ANN implementation with low-dimensional features of GoogLeNet and VGG-19.

Techniques	Type of Class	Sensitivity %	Precision %	Accuracy %	Specificity %	AUC %
ANN with features of GoogLeNet	colon_aca	97.2	95.35	95.1	99.41	98.74
	colon_bnt	95.84	96.89	94.2	98.98	97.82
	lung_aca	96.19	95.36	97.3	98.52	98.1
	lung_bnt	95.77	97.15	94.8	99.29	96.15
	lung_scc	97.43	96.22	96.1	99.38	97.56
	**average ratio**	**96.49**	**96.19**	**95.50**	**99.12**	**97.67**
ANN with features of VGG-19	colon_aca	96.54	92.25	97.1	98.15	97.4
	colon_bnt	91.33	96.95	90.8	98.64	98.23
	lung_aca	97.82	97.17	97.9	98.53	97.88
	lung_bnt	95.79	98.36	96.2	99.1	98.9
	lung_scc	97.94	96.19	97.6	97.98	97.67
	**average ratio**	**95.88**	**96.18**	**95.92**	**98.48**	**98.02**

**Table 3 bioengineering-10-00383-t003:** Results of implementing ANN with mixed features after and before applying PCA.

Techniques	Type of Class	Sensitivity %	Precision %	Accuracy %	Specificity %	AUC %
ANN with mixed features after PCA	colon_aca	100	100	100	100	100
	colon_bnt	99.1	100	99.5	100	100
	lung_aca	97.23	96.87	96.8	98.79	98.17
	lung_bnt	100	100	100	100	100
	lung_scc	97.41	96.95	97	99.2	99.74
	**average ratio**	**98.75**	**98.76**	**98.66**	**99.60**	**99.58**
ANN with mixed features before PCA	colon_aca	98.54	99.3	99.1	100	100
	colon_bnt	99.32	98.74	98.9	98.81	99.76
	lung_aca	98.2	98.65	98.3	100	97.88
	lung_bnt	98.18	98.12	98.3	99.42	99.62
	lung_scc	97.82	99.22	98.1	100	100
	**average ratio**	**98.41**	**98.81**	**98.54**	**99.65**	**99.45**

**Table 4 bioengineering-10-00383-t004:** Results of ANN implementation of fusion features for histological image diagnosis for the LC25000 dataset.

Techniques	Type of Class	Sensitivity %	Precision %	Accuracy %	Specificity %	AUC %
ANN with fusion features of GoogLeNet and handcrafted	colon_aca	98.88	100	99	100	100
	colon_bnt	99.11	98.35	98.9	99.27	99.72
	lung_aca	100	100	99.6	100	100
	lung_bnt	99.1	99.36	99	100	98.97
	lung_scc	100	100	99.6	100	100
	**average ratio**	**99.42**	**99.54**	**99.22**	**99.85**	**99.74**
ANN with fusion features of VGG-19 and handcrafted	colon_aca	100	100	99.7	100	100
	colon_bnt	99.25	100	99.4	100	100
	lung_aca	99.7	100	99.6	99.6	99.54
	lung_bnt	99.6	100	99.7	99.9	100
	lung_scc	99.6	100	99.8	100	99.78
	**average ratio**	**99.85**	**100.00**	**99.64**	**100.00**	**99.86**

**Table 5 bioengineering-10-00383-t005:** Results of ANN implementation of all systems in this study to diagnose histological images of the RC25000 dataset.

Techniques		Features	Colon_aca	Colon_bnt	Lung_aca	Lung_bnt	Lung_scc	Accuracy %
ANN		GoogLeNet	95.1	94.2	97.3	94.8	96.1	95.5
		VGG-19	97.1	90.8	97.9	96.2	97.6	95.9
ANN	Mixed features after PCA	GoogLeNet + VGG-19	100	99.5	96.8	100	97	98.7
	Mixed features before PCA	GoogLeNet + VGG-19	99.1	98.9	98.3	98.3	98.1	98.5
	Fusion features	GoogLeNet and handcrafted	99	98.9	99.6	99	99.6	99.2
		VGG-19 and handcrafted	99.7	99.4	99.6	99.7	99.8	99.6

**Table 6 bioengineering-10-00383-t006:** Comparison of the performance of the proposed system with the relevant existing systems.

Previous Studies	Accuracy %	Sensitivity %	Specificity %	AUC %	Precision %
Dabass, M. et al. [43]	97.4	97.2	97.1	-	97.21
Dabass, M. et al. [44]	95.85	-		-	-
Bukhari, S. et al. [45]	93.04	94.79	84.21	-	96.81
Masud et al. [15]	96.33	96.37	-	-	96.39
Liu, Y. et al. [46]	97.01	97.04	-	-	97.07
Attallah, O. et al. [47]	99.3	98.9	99.7	-	99
El-Ghany, S. et al. [48]	98.97	97.82	99.35	-	98.04
**Proposed model**	**99.64**	**99.85**	**100**	**99.86**	**100**

## Data Availability

The images applied to evaluate the performance of the systems were obtained from the LC25000 dataset, which is publicly available at the link: https://www.kaggle.com/datasets/andrewmvd/lung-and-colon-cancer-histopathological-images (accessed on 12 October 2022).

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
