# Peer review of "Histopathological Analysis for Detecting Lung and Colon Cancer Malignancies Using Hybrid Systems with Fused Features"

_bioengineering, 2023, doi:10.3390/bioengineering10030383_

Round 1
Reviewer 1 Report
In this manuscript, the authors investigated three different feature engineering methods to build deep learning predictive models for lung and colon cancer histopathological analysis. They have demonstrated that the use of fusion features of VCG-19 and hancrafted features resulted in highest predictivities. The manuscript has been well-organized and scientifically sound, which might benefit other researchers in similar fields. Thus, I will suggest acceptance of this paper.
Author Response
Thank you very much for your encouragement and acceptance of the paper in its present form.
Reviewer 2 Report
Authors should improve the writing of the manuscript.
Authors should include references for Artificial intelligence and Deep learning in the Introduction.
Materials and Methods should be more organized and concise, results should not be discussed in this section. Figure-2 can be moved to the result section.
A separate table should be incorporated for all the abbreviations used in the manuscript.
Authors should compare the performance of the final model with existing models by using their dataset.
Most importantly authors should provide the code for its reproducibility to support their claim.
Author Response
Responses are in the attached file.
Kindly be informed that all the changes and additions are highlighted in bright green for your kind reference.
|
Reviewer Comments |
Authors Response |
|
Authors should improve the writing of the manuscript. |
Thank you very much for your comment. The writing of the manuscript is improved |
|
Authors should include references for Artificial intelligence and Deep learning in the Introduction. |
Thank you very much for your comment. References 7, 8 and 9 about Artificial Intelligence and Deep Learning in Biomedical Image Diagnostics have been added to the Introduction section. |
|
Materials and Methods should be more organized and concise, results should not be discussed in this section. Figure-2 can be moved to the result section. |
Thank you very much for your comment. Figure 2 illustrates the performance of enhancement filters and is therefore placed in the same section of enhancement techniques. Section 4 presents the results of performance measures such as Accuracy, Sensitivity, Specificity, AUC, and Precision. Thus, transferring figure 2 to the results section will confuse the reader when the methodology of performing the filters in Section 3 and his figure in Section 4. |
|
A separate table should be incorporated for all the abbreviations used in the manuscript. |
Thank you very much for your comment.
A table has been added at the end of the manuscript to clarify the abbreviations |
|
Authors should compare the performance of the final model with existing models by using their dataset. |
Thank you very much for your comment.
Table 6 has been added, which contains a comparison between the performance results of our system with the systems of previous relevant studies. |
|
Most importantly authors should provide the code for its reproducibility to support their claim. |
Thank you very much for your comment.
Code and all requirements will be uploaded to the GitHub website in due course. |

Reviewer 3 Report
The manuscript tried to develop a hybrid model for cancer classification based on a published cancer imaging database. However, many key points regarding the input and structure of the model are missing or confusing. Some major problems include
1) The dimension of the last layer excluding output layer for VGG19 and GoogleLeNet are both 1*1*1000. But the manuscript states there are 2048 output features from both models. Where do these features come from?
2) The justification of using PCA after CNN model is not adequate. The features learned by CNN models are supposed to exclude non-significant and redundant ones in a non-linear manner. It's doubtful that performing PCA on the CNN learned features will improve the accuracy. Also, the paper it referred to (ref[29]) that uses 'feature merge after PCA' does not mention PCA at all.
3). The description of the ANN model after PCA is too simple and far from enough. I have no idea what the structure is except that it has 20 layers. Are these fully connected layers? How many nodes are there in each layer? Is the last layer softmax? Also, does it really need 20 layers? How do the authors justify it?
4). In Line 250, MSE stands for Mean Squared Error, not 'minimum quadratic error'.
5) How can you get a 99.7% accuracy when your sensitivity and specificity are both 100% in Table 4?
6). In Line 272-273, the sentence '... then combine them and send them to PCA separately' is really confusing. If you have combined the features, why and how do you send them to PCA separately?
The minor problems include
1). The statement 'so the dataset contains three types: malignant and two benign.' in line 170 is misleading and redundant. Actually there are 5 types of images including two subtypes of lung cancer, one type of colon cancer, one colon benign tissue and one lung benign tissue.
2). The notation of 5 types of images(e.g., lung_aca, colon_aca) is not consistent between Line 168-169 and figure 2(which makes _a subscript).
3). Please provide the full form for each abbreviation when it is mentioned for the first time in the manuscript. E.g., Contrast Limited Adaptive Histogram Equalization for CLAHE, Principle Component Analysis for PCA, etc.
In addition to the problems mentioned above, the algorithm itself is not novel. Also, no independent dataset is included for further validation. No comparison is made or mentioned by applying other deep learning models on the same dataset.
The authors really need to describe their model clearly before claiming it can get 99.64% accuracy in testing set for cancer classification.
Author Response
Responses are in the attached file.
Kindly be informed that all the changes and additions are highlighted in Turquoise for your kind reference.
|
Reviewer Comments |
Authors Response |
|
1) The dimension of the last layer excluding output layer for VGG19 and GoogleLeNet are both 1*1*1000. But the manuscript states there are 2048 output features from both models. Where do these features come from? |
Thank you very much for your comment. The fully connected layers that produce 1 * 1 * 1000 and these layers are classification layers have been deleted and replaced by ANN. A further explanation has been added as follows and included in the manuscript as section "3.4.ANN with Fusion Features of CNN".
The last layers of Googlenet and VGG-19 produce high-level features as follows: (7, 7, 512) and (7, 7, 512) for two models. The Global Average Pooling layer converts it from high -level features into a distinctive feature of 4096 features of both models. |
|
2) The justification of using PCA after CNN model is not adequate. The features learned by CNN models are supposed to exclude non-significant and redundant ones in a non-linear manner. It's doubtful that performing PCA on the CNN learned features will improve the accuracy. Also, the paper it referred to (ref[29]) that uses 'feature merge after PCA' does not mention PCA at all. |
Thank you very much for your comment. The PCA algorithm works by selecting the important features and removing the unimportant and redundant features effectively. References 32 and 33 are cited. A Hybrid Deep Transfer Learning of CNN-Based LR-PCA for Breast Lesion Diagnosis... 3D CNN-PCA: A deep-learning-based parameterization for complex geomodels |
|
3). The description of the ANN model after PCA is too simple and far from enough. I have no idea what the structure is except that it has 20 layers. Are these fully connected layers? How many nodes are there in each layer? Is the last layer softmax? Also, does it really need 20 layers? How do the authors justify it? |
Thank you very much for your comment. The fully connected layers of the GoogLeNet and VGG19 models have been replaced by an ANN that contains three layers: the input layer, the hidden layer, and the output layer. The number of hidden layers was adjusted through trial and error, as it was set to 10, 11, 12, and ..... 20 layers, and the results were recorded each time. ANN achieved the best performance when the hidden layers were 20 layers.
The ANN input layer consists of 455 input units based on the number of features. ANN contains 20 hidden layers connected to parameters (weights) at the layer level and other layers in which the required tasks are solved. The process is repeated from the first hidden layer to the last; the weights are adjusted, and the minimum square error (MSE) is calculated based on the difference between the actual and expected values​, as in Equation 4 [31]. The output layer sorts the features of each image with its appropriate class, which produces five neurons, each neuron representing class data set.
|
|
4). In Line 250, MSE stands for Mean Squared Error, not 'minimum quadratic error'. |
Thank you very much for your comment. Revised to Mean Squared Error". Thank you very much |
|
5) How can you get a 99.7% accuracy when your sensitivity and specificity are both 100% in Table 4? |
Thank you very much for your comment. Here the numbers are rounded by Excel cells. Edited; thanks a lot. "99.7% accuracy, 99.6% Sensitivity and 99.9% Specificity" |
|
6). In Line 272-273, the sentence '... then combine them and send them to PCA separately' is really confusing. If you have combined the features, why and how do you send them to PCA separately? |
Thank you very much for your comment. You are correct here "separately" word use is inappropriate, so omitted "separately" word. |
|
The minor problems include |
|
|
1). The statement 'so the dataset contains three types: malignant and two benign.' in line 170 is misleading and redundant. Actually there are 5 types of images including two subtypes of lung cancer, one type of colon cancer, one colon benign tissue and one lung benign tissue. |
Thank you very much for your comment.
The extra phrase has been removed as suggested |
|
2). The notation of 5 types of images(e.g., lung_aca, colon_aca) is not consistent between Line 168-169 and figure 2(which makes _a subscript). |
This is the names by dataset. A table has been added at the end of the manuscript to clarify the abbreviations. |
|
3). Please provide the full form for each abbreviation when it is mentioned for the first time in the manuscript. E.g., Contrast Limited Adaptive Histogram Equalization for CLAHE, Principle Component Analysis for PCA, etc. |
A table has been added at the end of the manuscript to clarify all abbreviations. |
|
In addition to the problems mentioned above, the algorithm itself is not novel. Also, no independent dataset is included for further validation. No comparison is made or mentioned by applying other deep learning models on the same dataset. |
Our systems can be generalized to a new data set and classified efficiently, and this is in future work. Table 6 has also been added to compare the performance of our systems with previous systems related to the same data set. |

Reviewer 4 Report
The authors describe an AI-based approach for diagnosing lung and colon cancer based on histological images.
The method works well, but the novelty is limited to the choice of the network architecture for feature extraction and the composition of the handcrafted features.Furthermore, existing methods have similar prediction qualities.
Please add/modify the following information:
The result is a prediction accuracy at SOTA level, but no further statement is made regarding the application as a diagnostic aid in clinical operation (e.g. inference time, compute requirements, ...) and thus no reference to diagnostic relevance is made. Also, the reproducibility of the results is not guaranteed by publicly accessible code on e.g. GitHub.
The choice of the MSE as a loss function should possibly be reviewed because the function normally does not fit to classification problems.
Some paragraphs can be significantly shortened.
The spelling should be re-checked. Some sentences in the abstract are incomplete.
Author Response
Responses are in the attached file.
|
Reviewer Comments |
Authors Response |
|
The result is a prediction accuracy at SOTA level, but no further statement is made regarding the application as a diagnostic aid in clinical operation (e.g. inference time, compute requirements, ...) and thus no reference to diagnostic relevance is made. Also, the reproducibility of the results is not guaranteed by publicly accessible code on e.g. GitHub. |
Thank you very much for your comment.
Prediction accuracy at SOTA level was not included in our study. |
|
The choice of the MSE as a loss function should possibly be reviewed because the function normally does not fit to classification problems. |
In the ANN network, it depends on reaching satisfactory accuracy by adjusting the weights. Each time the error rate between the actual and predicted values is measured through MSE, it is well suited to solve classification problems. |
|
Some paragraphs can be significantly shortened. |
The manuscript has been revised, and some paragraphs have been shortened. |
|
The spelling should be re-checked. Some sentences in the abstract are incomplete. |
The entire manuscript has been spell-checked |

Round 2
Reviewer 3 Report
Most major problems are still not resolved.